# Leaky Gum: The Revisited Origin of Systemic Diseases

**DOI:** 10.3390/cells11071079

**Published:** 2022-03-23

**Authors:** Do-Young Park, Jin Young Park, Dahye Lee, Inseong Hwang, Hye-Sung Kim

**Affiliations:** 1DOCSmedi Co., Ltd., 4F, 143, Gangseong-ro, Ilsanseo-gu, Goyang-si 10387, Korea; pdy@docsmedi.kr; 2Department of Gastrointestinal Endoscopy, Apple Tree Healthcare Center, 1450, Jungang-ro, Ilsanseo-gu, Goyang-si 10387, Korea; baadaak@appleden.com; 3Department of Orthodontics, Apple Tree Dental Hospital, 1450, Jungang-ro, Ilsanseo-gu, Goyang-si 10387, Korea; drdahaelee@appleden.com; 4Apple Tree Institute of Biomedical Science, Apple Tree Medical Foundation, 1450, Jungang-ro, Ilsanseo-gu, Goyang-si 10387, Korea

**Keywords:** biofilm, gingival sulcus, junctional epithelium, leaky gum, leaky gut, mucosal barrier, oral microbiome, systemic disease

## Abstract

The oral cavity is the gateway for microorganisms into your body where they disseminate not only to the directly connected respiratory and digestive tracts but also to the many remote organs. Oral microbiota, travelling to the end of the intestine and circulating in our bodies through blood vessels, not only affect a gut microbiome profile but also lead to many systemic diseases. By gathering information accumulated from the era of focal infection theory to the age of revolution in microbiome research, we propose a pivotal role of “leaky gum”, as an analogy of “leaky gut”, to underscore the importance of the oral cavity in systemic health. The oral cavity has unique structures, the gingival sulcus (GS) and the junctional epithelium (JE) below the GS, which are rarely found anywhere else in our body. The JE is attached to the tooth enamel and cementum by hemidesmosome (HD), which is structurally weaker than desmosome and is, thus, vulnerable to microbial infiltration. In the GS, microbial biofilms can build up for life, unlike the biofilms on the skin and intestinal mucosa that fall off by the natural process. Thus, we emphasize that the GS and the JE are the weakest leaky point for microbes to invade the human body, making the leaky gum just as important as, or even more important than, the leaky gut.

## 1. Introduction

Humans internalize the microbiota of this planet through the oral cavity, either temporarily or permanently. The oral cavity harbours the second most abundant microorganisms after the gastrointestinal (GI) tract in a variety of distinct habitats, such as teeth, tongue, gingival sulcus (GS), palate, saliva, buccal mucosa, and throat. The expanded Human Oral Microbiome Database (eHOMD v3, http://homd.org, accessed on 16 February 2022) established during the Human Microbiome Project enlists at least 774 microbial species to date.

As the old dogma that the lungs and placenta are sterile becomes obsolete [1,2,3,4,5], the oral microbiota has proven to be the primary source of the bacterial microbiota in human organs [6]. For one, microaspiration during respiratory activity, such as oral breathing, affects the lung microbiota [7]. In addition, dietary patterns dynamically affect the microbiome profile of the GI tract either by microbial contamination or by supplying specific nutrients for microbial commensals, even manipulating the pathophysiology of cancerous diseases [8,9] as well as regulating immune responses across the gut–brain axis [10,11]. As such, along with the revolution of human microbiome research, much effort has been dedicated to figuring out the relationship between the oral and gut microbiota, which has been dubbed the “oral–gut–brain axis” [12,13,14,15,16,17].

The microbiota in the gut seek opportunities to breach the dysfunctional gut mucosal barrier to reach the underlying immune system, resulting in “leaky gut” syndrome. This type of leak occurs through the cellular junctions between the intestinal epithelia. In the oral cavity, the microbes can easily colonize on the surface of the hard tissue of the tooth enamel to form biofilm and remain sclerotized until proper interventions come in. The growing body of the biofilm not only acts as a wedge disjoining the tooth and the gingiva, enlarging the GS depth but also deploy an ample number of microbes near the sulcus epithelium, enhancing the opportunity for microbial infiltration into the oral mucosa [18]. Thus, microbial infection in the oral cavity is accelerated by both physical and biological processes.

In this review, we gather current knowledge of disease-related oral pathogens and contrast the anatomical structures of oral versus gut mucosal layers in the context of microbial leak into a human body, embossing the role of oral pathogens in the development of systemic diseases.

## 2. Hyperpermeable Intestine—Leaky Gut

The largest portion of research funds has been digested by gut bacteria because the majority of human microbiota resides in the colon [19]. The findings of their roles in human immune systems have been extensively illuminating. The human intestine is the widest and longest space in contact with microorganisms compared to the oral cavity or the skin. It boasts almost 10 metres of length and 400 m^2^ of luminal surface area. In addition, it allows the passage of about 60 tons of food during a lifetime while processing digestion and absorption, making the role of gut bacteria even more important [20]. Especially, gut bacteria, numbering almost equivalent to the human cells [19], metabolize dietary fibers to yield short-chain fatty acids, an essential task of which humans are not capable. They not only provide intestinal cells with immune substances and vitamins but also keep the intestinal homeostasis. The homeostasis of the intestinal microbiome itself and between the intestinal microbiome and the host has been conceptualized as a “symbiosis” of the intestinal ecosystem [21].

Perturbation factors, such as stress, smoking, alcohol consumption, eating processed foods, and overuse of antibiotics, have a certain effect on the ecosystem of the intestinal microbiome. The disturbance is usually absorbed by the resilience of the intestinal ecosystem, but repeated exposures to such risk factors would lead to “dysbiosis”, a continuous status of imbalance between gut microbiota and their host [22]. For example, when antibiotics deplete intestinal bacteria who are responsible for converting primary bile acids into secondary bile acids, such as deoxycholate, lithocholate, ursodeoxycholate, hyodeoxycholate, and ω-muricholate, the resistance to *Clostridium difficile* decreases [23], resulting in pseudomembranous colitis and persistent diarrhoea that claim lives of tens of thousands of people in the US [24,25]. Likewise, it is now well-accepted that a dysbiosis of gut microbiome can affect not only inflammatory bowel disease (IBD), constipation, indigestion, and obesity but also the occurrence and prognosis of hypertension, diabetes, cancer, and cardiovascular diseases (CVD) [26,27,28,29,30,31,32,33]. 

During the golden era of gut microbiome research, the term “leaky gut”, previously proposed and used in the field of alternative medicine and dietetics, was revisited as a compatible explanation of “increased intestinal permeability” (Figure 1a,b) [34,35,36,37]. The rationale behind the term lies in the concurrent pathogenesis of intestinal and systemic diseases caused by the infiltration of enteric bacteria and virulence factors into the intestinal mucosal membrane when the epithelial barrier function is disrupted [36,38,39]. The intestinal hyperpermeability has often been accompanied by the changes of tight junction proteins in the epithelium or the increased bacterial endotoxin in the bloodstream, endotoxemia [36]. For instance, patients with IBD, irritable bowel syndrome (IBS), liver diseases, pancreatitis, diabetes, chronic heart failure, depression, and other chronic diseases often exhibit increased permeability and epithelial barrier dysfunction [40,41,42]. As evidence builds up and interests from diverse research fields expand further, the methodologies for the measurement of intestinal barrier function have been extensively developed [42].

The association between intestinal and systemic diseases can be found in extra-intestinal manifestations (EIMs) of intestinal diseases. For example, the EIMs of IBD affect joints, eyes, liver, lung, and pancreas [43]. About 15% of people with ulcerative colitis (UC) [44] and up to 40% of patients with Crohn’s disease [45,46,47,48], both are types of IBD, have skin issues. In some EIMs cases, such as peripheral arthritis, oral aphthous ulcers, episcleritis, and erythema nodosum, symptoms can improve on standard treatment of the intestinal inflammation [43]. In addition, IBD and periodontitis have been reported to affect each other, and several nutritional deficiencies and systemic diseases are known to be manifested in the oral cavity [49,50], which is supporting bidirectional influence in the context of an oral–gut axis. In the same vein, bacteria can travel via the bloodstream to reach and colonize in the tumour microenvironment (TME) of melanoma, lung, ovarian, glioblastoma, pancreatic, bone, and breast cancer where ~10^6^ intratumoural bacteria per palpable 1-cm^3^ tumour can be found [32,51].

## 3. Gum and Gut Mucosal Barriers

The lumen of the digestive tract, a twisted hollow tube from the oral cavity to the anus, is continuously overloaded with the external environment [52]. Thus, just as the human skin protects our body, the oral and intestinal mucosa, which cover the inner surface of our body, should exert barrier function physically and physiologically [38,39]. Yet unlike the skin whose major function is building a physical barrier, the major function of the mucosal membrane comprises several physiological barriers. For example, saliva and mucus on the epithelium contain antibacterial substances, such as lactoferrin, lysozyme, commensal flora, and antibacterial peptides, which inhibit pathogen colonization [53,54]. 

The surface layers of the skin (epidermis), the oral cavity, and the gastrointestinal tract share both similar and dissimilar structures and functions. The epidermis comprises several cellular layers strongly bound with intercellular junctions of which the surface is covered by the stratum corneum [55]. The intestinal epithelial layer is also filled with cells interconnected with strong intercellular junctions (Table 1). The luminal side of the intestinal epithelial layer, however, is made up of a thinner monolayer, is supported by the connective tissue underneath only, and has no stratum corneum covering the layer. Thus, from an anatomical point of view, the intestinal mucosa is less tolerable to an environmental shift such as dysbiosis of a microbial community, which leads to bacteremia or endotoxemia through the increased intestinal permeability [56]. To compensate for the weakness of the epithelial layer, the intestinal mucosa always covers the epithelium with mucus that not only acts as a lubricant between the mucosa and the luminal passengers but also contains a lot of antibacterial substances, indicating that the gut mucosa has both chemical and biological barrier functions [57]. Pathogenic intruders who survive in the mucosal layer and leak through the epithelial barriers face mucosal-associated lymphoid tissue (MALT) that takes up ~70% of the entire immune system of a human body (Figure 1b). In addition, ~80% of plasma cells residing in gut-associated lymphoid tissue (GALT), a major part of MALT, wage a deliberate war against antigens originating from ~4 × 10^13^ commensal microbes and more than 30 kg of food proteins yearly [19,52,58]. Consequently, the gastrointestinal system plays a pivotal role in immune surveillance.

Mucosal immunity in the oral cavity is also gaining traction with the advance of microbiome research [59,60]. The oral mucosa, like the skin, is composed of both keratinized and non-keratinized tissues (Figure 1c). The thickness of the keratinized layers of the oral mucosa, however, is thinner than that of the skin. Thus, the oral mucosa, similar to the intestinal mucosa, supplements chemical and biological defence functions using saliva. The oral mucosa also consists of 3–5 cellular layers that are thicker than the intestinal mucosa monolayer. Anatomically and histologically, the oral mucosa appears as a transitional layer between the intestinal mucosa and the skin [61].

## 4. Gingival Sulcus and Junctional Epithelium

For humans, the primary teeth erupt through the mucous membrane from the inner alveolar bone at about six months of age, forming unique structures originating from the interface between the exposed teeth and the surrounding tissues. Those interface structures, the GS and the JE below the GS, are essential for the survival of animals that need mastication of ingested food (Figure 1d). For adults with permanent teeth, the healthy GS depth can reach 3 mm. Thus, below the GS there should be a sealing layer that binds the soft tissue, especially the JE, with the surface layer of the hard tissue (enamel and cementum), protecting the tissues from external challenges. The JE has the highest turnover rate (4–6 days) of all oral epithelia and remains undifferentiated and non-keratinized [62]. In the JE, the binding proteins are generated by the basal layer of soft tissues only to form HD with the hard tissues of a tooth (Figure 1d, magnified box). The internal basal lamina (IBL), the intercellular space in the HD, are relatively wide, allowing water-soluble substances to pass through them with ease. These structural limitations of the HD between the JE and the tooth can provide pathogens with a good opportunity to invade the human body [63,64,65]. To compensate for this inherent structural weakness, immune cells such as polymorphic nuclear leukocytes (PMNs) transude into the GS together with gingival crevicular fluid (GCF), taking constant vigilance even without any signs of inflammation [66,67]. Ironically, to allow immune cells to pass through a JE layer, the JE cells have fewer desmosomes that bind the cells vertically than the other oral epithelia, adding another structural instability to JE [68].

The GS provides the perfect space for biofilm accumulation. Biofilms on the skin, oral mucosa, and gut mucosa are washed out along with hygiene activities, digestive processes, exfoliation, and defecation. The surface layer of the tooth enamel, however, does not fall off because it lacks cell division and maintains its structure unless external physical and biological intervention is applied. Thus, if the GS is not properly managed, biofilms will inevitably accumulate during lifetime [69] even to the level of thickness enough to ward off antibiotics [70] (Figure 2a,b). The biofilm accumulation induces inflammatory responses that erode alveolar bone and increases the GS depth, resulting in the formation of the periodontal pocket (PP). The deepened PP in turn makes it difficult to remove the biofilm in the PP. This vicious cycle results in increased inflammatory reactions, i.e., periodontal diseases [18,71]. 

The important roles of the GS and the JE were embossed in a seminal study conducted on 417 patients at 11 nursing homes in Japan [72]. In this study, older patients who received oral care exhibited lower cases of pneumonia, febrile days, and death from pneumonia, while showing improved metrics of activities of daily living (ADL) and cognitive functions evaluated with Mini-Mental State Examination (MMSE). By contrast, the total mortality rate was greater in the dentate group (13.5%, 28/208) than in the edentate group (10%, 16/158). The mortality rate of dentate and edentate groups with oral care was similar (6% and 7%, respectively), but without oral care, the mortality rate of the dentate group (20%) was higher than that of the edentate group (13%) even if ADL and MMSE scores were slightly worse in the edentate group at the time of final evaluation. The reduced mortality rate in the edentate group without oral care, although not fully discussed in this study, may indicate that the edentate state is somehow advantageous for longevity if proper oral care cannot be administered. Indeed, the spot where the tooth is removed becomes covered with mucosal membrane and transforms like the mucosa of the skin and the intestine (Figure 2c). In other words, the absence of teeth may render more effective protection from bacterial infections by removing the vulnerable structures originating from the GS and the JE [73].

## 5. Focal Infection Theory and Leaky Gum

Concerns have already existed since the end of the 19th century that the oral cavity could be a source of human pathogenic microbes. In the 1890s, Willoughby D. Miller, who studied at the Koch Institute, warned of the dangers of oral microbes [74,75]. Miller, riding on the bandwagon of the “germ theory” of disease established at the end of the 19th century, suggested that the oral cavity, a breeding ground for many pathogens, could be an origin of many diseases of unknown aetiology, such as CVDs, pneumonia, angina pectoris, and foot gangrene [76]. Miller’s study has established a modern daily routine for oral care, such as brushing teeth three times a day and flossing. His argument was later accepted as “oral sepsis” by British surgeon William Hunter in the early 1900s [77] and expanded as “focal infection theory” by American physician Frank Billings in the 1910s [78]. It was further amplified by Henry Cotton who claimed that mental illness could be improved by tooth extraction or tonsillectomy [79]. Even accepted by the physicians at Johns Hopkins University and Mayo Clinic, the theory was implemented into routine clinical practice. The theory was so widely expanded that Russell Cecil, an eminent author of Cecil Essentials of Medicine, also joined the club. In the 1940s, however, Hobart A. Reinmann and W. Paul Havens pointed out, in their critical appraisal of focal infection in relation to systemic disease, that the theory lacks clinical evidence and the causative relationship of infections of teeth and tonsils to systemic disease is unproven [80]. Consequently, in the late 1950s, the theory gradually vanished and was regarded as fringe medicine. 

In the 21st century, the focal infection theory began to be revisited from a different perspective [81]. For example, bacteremia, a temporary infiltration of bacteria into blood vessels, has been regarded as an illness resolved by immune responses within an hour [65]. Recent culture-independent microbial research techniques, however, have shown that bacteria or bacterial DNA are always present in the blood vessels of healthy people [82,83,84]. These findings suggest that bacteremia may not be a temporary nor a localized problem. Furthermore, it has been revealed that microbes can be found in the lungs of healthy people [5,85,86] and cancer patients [87], the placenta of healthy pregnant women, albeit controversial [1,2,3,4], and the brains in Alzheimer’s disease (AD) [88,89,90,91,92], which had long been considered sterile.

## 6. Oral Pathogens and Systemic Diseases

The origin of bacteria found in remote organs converges to the oral cavity [93,94,95,96] (Figure 3). For example, the placental microbiome profiles were most comparable to those of the oral microbiome [1]. The overlap of the unique members of oral microbes with other remote organs is consistent with previous clinical studies in which *Fusobacterium nucleatum*, a Gram-negative oral anaerobe, were clinically suspected to be a major risk factor in colorectal cancer [97,98,99], oral squamous cell carcinoma (OSCC) [100], and in preterm and term stillbirth [101,102]. Likewise, an infamous oral pathogenic bacterium, *Porphyromonas gingivalis*, is related to pancreatic cancer [103], colorectal cancer [104,105,106], liver health [107], rheumatoid arthritis [108,109], diabetes [110,111,112], OSCC [113,114], and neurodegenerative diseases [88,89,90,91,92,115,116,117,118]. In the case of atherosclerotic CVD, when the vascular tissues of the coronary and femoral arteries of the patients with CVD were examined, *P. gingivalis* was found in 42 out of 42 patients [119]. Thus, previously explained by the passive accumulation of fat, the aetiology of CVD is now leaning toward inflammatory responses of the vascular endothelium [120,121]. Notably, live *P. gingivalis* is known to traffic into endoplasmic reticulum-rich autophagosomes [122] and target host ectonucleotidase-CD73 [123] for its chronic survival, replication, and persistence in the dysbiotic human gingival epithelia [124].

Recently, Kitamoto et al. demonstrated the mechanistic underpinnings by which periodontal inflammation due to oral infection contributes to the pathogenesis of extra-oral diseases [12]. In this elaborated study using mice, they showed that periodontitis aggravates gut inflammation by translocating oral *Klebsiella/Enterobacter* species to the lower digestive tract where it colonizes ectopically to elicit colitis through IL-1β. In parallel, oral Th17 cells induced by oral pathobiont expansion migrate to the gut and promote colitis, constituting both microbial and immunological pathways that link oral and gut health. Furthermore, Dong et al. showed that the alterations of the oral microbiota, especially *F. nucleatum* colonization in CRC locus, can change the gut bacterial composition within tumours and influence the therapeutic efficacy and prognosis of radiotherapy for primary rectal cancer and CRC liver metastases in mouse models [125]. Kartal et al. applied shotgun metagenomic and 16S rRNA sequencing to faecal and saliva samples from pancreatic ductal adenocarcinoma (PDAC) to identify diagnostic classifiers [96]. While proving that the faecal metagenomic classifiers outperformed the saliva-based classifiers, they also confirmed that the strains of faecal PDAC-associated microbes originate from the oral cavity. Thus, the growing body of examples that show the close relationship of oral pathogens with a variety of systemic diseases enabled the introduction of the term “periodontal medicine”, to describe how periodontal infection and inflammation affect extraoral illness [126,127]. As such, the oral cavity needs to be re-evaluated as a more pivotal organ with the revolution of microbiology in the 21st century [71].

There are many disease model systems proving that the oral bacteria induce systemic diseases, such as CVD, type II diabetes mellitus (T2DM), OSCC, and AD (Table 2). For one, the infection of aortic lesions with *P. gingivalis* activates adhesion molecules such as intercellular adhesion molecule 1 (ICAM-1) and vascular cell adhesion molecule 1 (VCAM-1), leading to chronic inflammation via migration of more immune cells to the lesion sites [128]. The microarray analysis demonstrated that *P. gingivalis*-treated human aortic endothelial cells (HAECs) upregulated expression levels of ICAM-1, VCAM-1, and interleukin-6 (IL-6). As well as ICAM-1 and VCAM-1 upregulation, pathological enlargement of atherosclerotic lesion area were well demonstrated in hyperlipidemic (*Apoe*^−/−^) mice orally challenged with *P. gingivalis* [128,129]. As an effective molecule, lipopolysaccharide from *P. gingivalis* (PgLPS) was established to promote inflammatory response as increasing mononuclear cell adhesion to human umbilical vein endothelial cells (HUVECs) via ICAM-1 and Toll-like receptor 2 (TLR-2)-dependent mechanism [130]. Subcutaneous infection of obese pigs with *P. gingivalis* also showed enhanced aortic and coronary arterial atherosclerosis [63]. In addition to *P. gingivalis*, intravenous infection of hyperlipidemic mice with *Aggregatibacter actinomycetemcomitans* can promote and accelerate atherosclerotic plaques [131] and time-dependently elevate matrix metalloproteinase-9 (MMP-9) expression [132]. The MMP-9, derived from macrophage, has been highlighted as a risk factor of acute atherosclerosis due to its proteolytic activity of advanced atherosclerotic plaque rupture [133].

T2DM is a highly prevalent metabolic disease characterized by prolonged high glucose levels in the blood. Insulin resistance on peripheral tissues has been focused on as the major causing factor of T2DM [144]. Recently, many microbiologists designated gut dysbiosis as a critical factor of insulin resistance development in T2DM accompanied by gut barrier dysfunction [145]. Interestingly, oral infection of mice with *P. gingivalis* can also induce gut dysbiosis, leading to insulin resistance via a pathway through endotoxin entrance and chronic inflammation [134]. Mice pre-treated with *P. gingivalis*, *F. nucleatum*, and *Prevotella intermedia* showed accelerated insulin resistance after three months of high-fat diet (HFD) feeding [135]. The branched-chain amino acid (BCAA) biosynthesis activity of *P. gingivalis* is a suggested mechanism of insulin resistance development, as evident that BCAA aminotransferase-deficient (∆*bcat*) *P. gingivalis* strain can neither induce insulin resistance nor upregulate serum BCAA in HFD mice model [136]. 

OSCC is the most malignant cancer of the oral cavity with an increasing rate of incidence, and the risk factors for OSCC include alcohol consumption, smoke, and human papillomavirus [146]. Interestingly, two independent groups suggested that *P. gingivalis* administration can significantly increase the number and diameter of the lesions in tongue tissues of mice pre-treated with carcinogen 4-nitroquinoline-1 oxide (4NQO). They provided two pathways, dysregulated lipid metabolism and CD11b^+^ myeloid-derived suppressor cells (MDSCs) infiltration, involved with OSCC deterioration by the pathogen [113,114]. Indeed, abnormal lipid regulation by increased expressions of fatty acid-binding protein 4 (FABP4) and FABP5 has been reported to have a crucial role in OSCC development via activation of mitogen-activated protein kinase (MAPK) pathway and MMP-9 [147,148]. By contrast, CD11b^+^ is responsible for MDSCs migration to tumour microenvironment where the cells have an immunosuppressive role that favours tumour progression [149].

AD is one of the representative neurodegenerative diseases diagnosed with senile plaques and abundant neurofibrillary tangles, which can be deteriorated by oral pathogenic infection. Gingival-infected *P. gingivalis* was reported to exacerbate the accumulation of Aβ plaques and inflammatory cytokines in brain specimens of amyloid precursor protein (APP) transgenic mice [140]. Interestingly, the anatomic analysis demonstrated that *P. gingivalis* genomic DNA was detected in brain specimens of 9 out of 12 *Apoe*^−/−^ mice orally challenged with *P. gingivalis* for 24 weeks [139], implicating that *P. gingivalis* can penetrate the gum and enter the blood-brain barrier (BBB). The result that intravenous injection of *P. gingivalis* into rats enhanced tau hyperphosphorylation in the hippocampus can reinforce the theory of BBB penetration of *P. gingivalis* [143]. As similar atherosclerotic CVD, PgLPS has also been designated as *P. gingivalis*-driven virulence factor affecting AD. It was reported that PgLPS treatment to rat brain neonatal microglial cells promoted the release of inflammatory mediators such as TNF-α and IL-6 [138], and palatal gingival infection of PgLPS into rats induced alveolar bone loss and increased serum Aβ levels [142]. Middle-aged wild-type (WT) mice intraperitoneally challenged with PgLPS for 5 weeks represented learning and memory deficit and microglia-mediated neuroinflammation, although age-matched mice deficient for cathepsin B (*Catb*^−/−^) were insensitive to PgLPS [141]. In addition to PgLPS, gingipain is an AD virulence factor, a unique class of cysteine proteinase that comprises Lys-gingipain (Kgp) and Arg-gingipain (Rgp). The modulatory role of the gingipain on neuroinflammation was well-established using Kgp and Rgp inhibitors (KYT1 and KYT36, respectively) or *P. gingivalis* KDP129, a gingipain-deficient mutant strain [137]. In this study, KYT1 and KYT36 treatment effectively inhibited *P. gingivalis*-driven increased expression of IL-6 and TNF-α in immortalized mouse microglial cell line MG6. Injection of *P. gingivalis*, but not KDP129 strain, into the somatosensory cortex of mice can recruit microglia to the injection site, revealing that gingipain is the effective factor for microglial migration and accumulation around *P. gingivalis* in the brain [137].

## 7. Conclusions and Perspectives

Thanks to rapid advances in gene sequencing technology combined with nanotechnology and information technology, the human microbiome proves to be present even in bodily sites previously known to be sterile. Intriguingly, the microbiome inside the human body mostly originates from the oral cavity, reminiscing the focal infection theory backed by more recent scientific proofs. Indeed, the oral cavity has unique mucosal structures such as the PP and the JE with innate vulnerability where oral pathogens can colonize for life and leak into blood vessels to circulate throughout the body, resulting in many systemic diseases in remote sites. As of now, although just a few molecules (PgLPS or gingipain) have been unravelled as oral pathogen-induced toxic molecules, we need to figure out many other causative candidates derived from oral pathogens to reinforce the attribution of leaky gum to various types of systemic diseases. Filled with anticipation for more causative evidence from well-designed empirical studies, we also need to focus on how to provide a leaky gum with a protective shield made of biological, not physicochemical, knowledge. By doing so, we can look forward to the realization of more prominent personalized medicine for systemic health by striking a balance between oral microbiota and its host.

## Figures and Tables

**Figure 1 cells-11-01079-f001:**
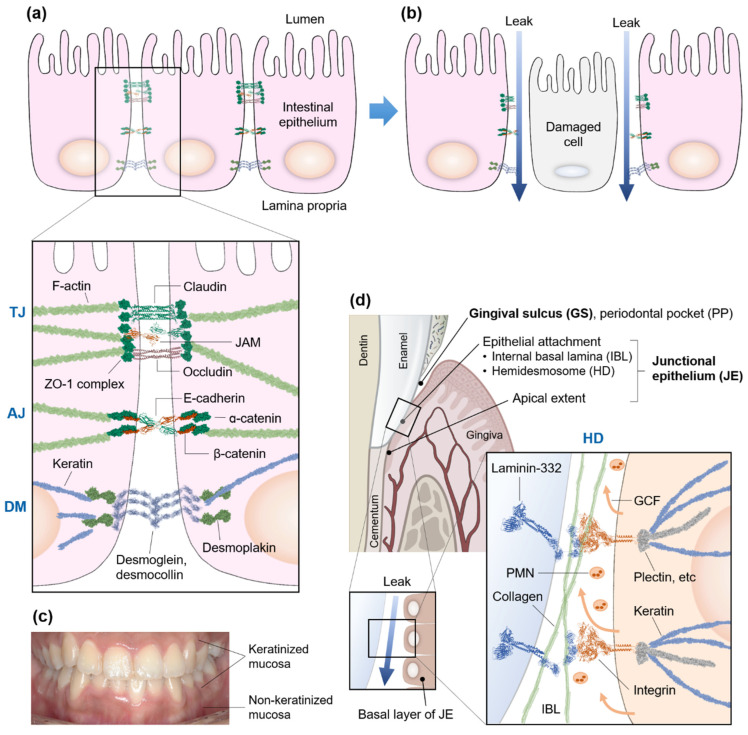
Schematics of differences in the cellular junctions between the intestinal and oral epithelia. (**a**) The intestinal epithelia are interconnected and communicate with each other through junctions, such as tight junction (TJ), adherence junction (AJ), desmosome (DM), and gap junction (GJ, not shown). (**b**) When these barriers are disrupted because of epithelial damages, pathogens, and chemicals in the luminal side can leak through the damaged cellular gaps into the lamina propria (blue arrows) whereby the MALT implements immune responses, resulting in a leaky gut syndrome. (**c**) Keratinized and non-keratinized oral mucosa. (**d**) Unlike the intestinal leakage through the cell-to-cell junctions, the leakage in the oral mucosa occurs through the hemidesmosome (HD) between the basal layer of the junctional epithelium (JE) and the hard surface layer of a tooth, which is inevitably and more frequently exposed to the physical and biological challenges. The internal basal lamina (IBL), an HD interface, is inhabited with collagens and binding proteins, such as laminin-332 and integrin. The periodontal pocket (PP), a pathologically deepened gingival sulcus (GS), occurs with the detachment of the connective tissues of the gingiva from the tooth surface. The JE below the GS is ~0.15 mm wide and 1–2 mm high, remains non-keratinized and undifferentiated, and has the highest turnover rate (4–6 days) of all oral epithelia. The polymorphonuclear leukocytes (PMNs) are also secreted with gingival crevicular fluid (GCF) from the basal layer to keep a lookout for any hostile intruders. ZO-1: zonula occludens-1, JAM: junctional adhesion molecule.

**Figure 2 cells-11-01079-f002:**
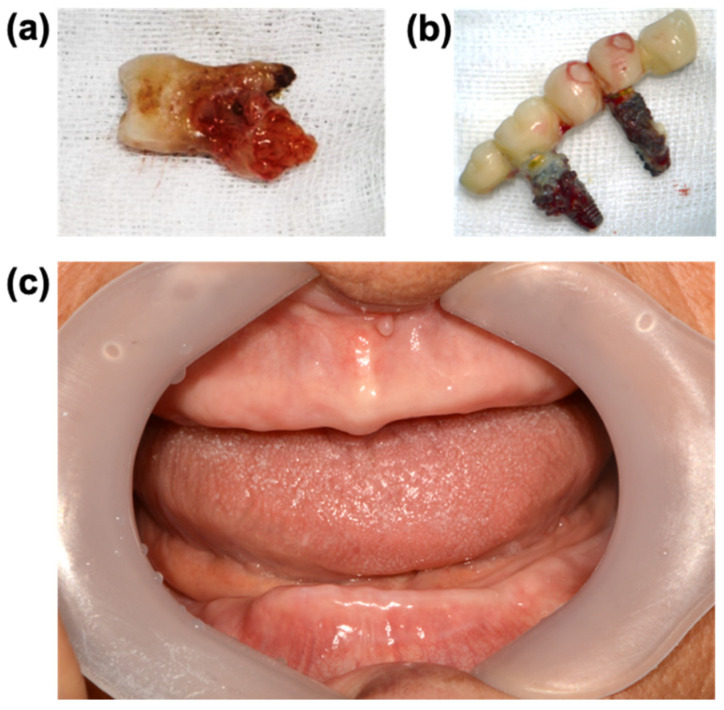
Biofilms built on (**a**) a tooth’s surface and (**b**) extracted implants. The hard surface of a tooth root, an implant, and a crown prosthesis abutting an implant shaft provide a solid ground on which biofilms can accumulate for a lifetime if not well cared for. (**c**) The edentate oral cavity. The toothless oral mucosa is free of the GS and the JE, making it less vulnerable to infection.

**Figure 3 cells-11-01079-f003:**
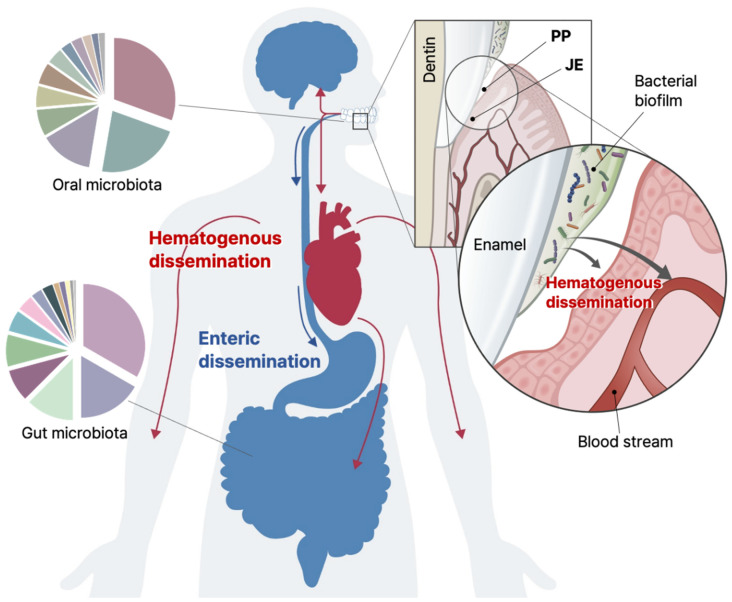
The oral cavity as the origin of the internal microbiome in humans. The microbiome in the oral cavity can disseminate to the remote sites of the body, such as the brain, stomach, intestines, and heart, via hematogenous and enteric pathways. The PP, a pathologically deepened GS due to microbial infection and colonization, gradually allows detachment of the connective tissues of the JE from the tooth surface. The epithelial layer of the apical JE is thin enough for bacterial virulence factors as well as pathogenic bacteria, such as *P. gingivalis*, to infiltrate into the bloodstream, resulting in a leaky gum syndrome. The microbiome of the oral cavity also affects gut microbiome profiles by moving through the gastrointestinal tract, causing a variety of gut-related diseases, such as IBD, IBS, and colon cancer.

**Table 1 cells-11-01079-t001:** Comparison of skin, oral mucosa, and intestinal mucosa.

Epithelium		Skin	Oral	Intestinal
KERATINIZED TISSUE		Exist	Partially exist	Not exist
EPITHELIAL LAYER		Multiple layers	Multiple layers	Single layer
INTERCELLULAR JUNCTIONS	Tight junction	Exist	Exist	Exist
	Adherence junction	Exist	Exist	Exist
	Desmosome	Exist	Exist	Exist
	Gap junction	Exist	Exist	Exist
	Hemidesomosome	Not exist	Exist	Not exist

**Table 2 cells-11-01079-t002:** The systemic disease models induced by oral pathogens.

		Oral Pathogens	Models	Infection Methods	Experimental Results	Year	Ref.
Atherosclerotic CVD	In vitro	*P. gingivalis* 381	HAECs	6 h infection	Increased ICAM-1, VCAM-1, and IL-6 expression	2005	[128]
		*P. gingivalis* ATCC33277-driven PgLPS	HUVECs	24 h infection	Increased adhesion of mononuclear cells to HUVECs via ICAM-1 and TLR-2 dependent mechanism.	2008	[130]
	In vivo	*P. gingivalis* 381	*Apoe*^−/−^ mice Infected (*n* = 25) Non-infected (*n* = 25)	Oral infection 5 times per week over 3 weeks	Increased aortic atherosclerosis.	2003	[129]
		*P. gingivalis* 381	*Apoe*^−/−^ mice Infected (*n* = 6) Non-infected (*n* = 6)	Oral infection 5 times per week over 3 weeks	Increased aortic ICAM-1, VCAM-1 immunostaining.	2005	[128]
		*P. gingivalis* 381 or A7436	Pigs Infected (*n* = 23) Non-infected (*n* = 13)	Subcutaneously infection 3 times per week for 5 months	Increased aortic and coronary arterial atherosclerosis.	2005	[63]
		*A. actinomycetemcomitans* AT445b	*Apoe*^−/−^ mice Infected (*n* = 9) Non-infected (*n* = 9)	Intravenous infection once a week for 4, 6, or 8 weeks	Increased aortic MMP-9 expression and serum CRP.	2008	[132]
		*A. actinomycetemcomitans* HK1651	*Apoe^shl^* mice Infected (*n* = 6) Non-infected (*n* = 6)	Intravenous infection 3 times per week over 3 weeks	Increased atherosclerotic plaque, serum C-reactive protein (CRP), IL-6, and aortic ICAM-1.	2014	[134]
T2DM	In vivo	*P. gingivalis* W83	Mice	Oral infection twice per week for 5 weeks	Increased gut dysbiosis, gut barrier invasion, serum endotoxin, insulin resistance.	2014	[134]
		*P. gingivalis* ATCC33277, *F. nucleatum*, *P. intermedia*	Mice Infected (*n* = 16) Non-infected (*n* = 13)	Oral infection 4 times a week for 4 weeks, thereafter normal diet or HFD-fed for additional 3 months	Increased periodontal dysbiosis, insulin resistance in HFD-fed mice.	2017	[135]
		*P. gingivalis* ATCC33277 (WT) or *∆bcat*	Mice WT infected (*n* = 6) ∆*bcat* infected (*n* = 6) Non-infected (*n* = 6)	Oral infection twice per week for 4 weeks concomitantly HFD-fed	*P. gingivalis* (∆*bcat*) cannot induce insulin resistance in HFD-fed mice.	2020	[136]
OSCC	In vivo	*P. gingivalis* 381	Mice Infected (*n* = 15) 4NQO-treated (*n* = 20) 4NQO-treated + infected (*n* = 20) Control (*n* = 10)	4NQO treatment for 8 weeks, thereafter oral infection with *P. gingivalis* for 8 weeks	Enhanced OSCC induction and dysregulated lipid metabolism in 4NQO-treated mice.	2018	[113]
		*P. gingivalis* ATCC33277	Mice 4NQO-treated + infected (*n* = 12) Non-infected (*n* = 6)	4NQO treatment for 16 weeks, thereafter oral infection with *P. gingivalis* for 10 weeks	Enhanced OSCC induction and increased infiltration of CD11b^+^ MDSCs in 4NQO-treated mice.	2020	[114]
AD	In vitro	*P. gingivalis* ATCC33277	Immortalized mouse microglial cell line MG6	3, 6, or 12 h infection of *P. gingivalis* in the presence and absence of KYT1 (Rgp inhibitor) and KYT36 (Kgp inhibitor)	Increased expression levels of IL-6 and TNF-α, which was inhibited by KYT1 and KYT36 treatment.	2017	[137]
		PgLPS	Rat brain neonatal microglia	18 h infection	Activated microglial release of cytokine TNF-α, IL-6, and MMP-9.	2020	[138]
	In vivo	*P. gingivalis* 381, *Treponema denticola* ATCC 35404, *Tannerella forsythia* ATCC 43037, and *F. nucleatum* ATCC 49256	*Apoe*^−/−^ mice Mono-infected (*n* = 12) Multi-infected (*n* = 12) Non-infected (*n* = 12)	Oral infection for 24 weeks	*P. gingivalis* genomic DNA was detected in mice brain (9 out of 12 at 24 weeks).	2015	[139]
		*P. gingivalis* ATCC33277	APP transgenic mice Infected (*n* = 14) Non-infected (*n* = 12)	Gingival infection	Exacerbated Aβ plaques and inflammatory cytokines in the brain of AD mouse model.	2017	[140]
		PgLPS	Mice Young WT mice (2 months, *n* = 6) Middle-aged WT mice (12 months, *n* = 6) Young *Catb*^−/−^ mice (*n* = 6) Middle-aged *Catb*^−/−^ mice (*n* = 6)	Intraperitoneal infection daily for 5 weeks	PgLPS induced learning and memory deficit in middle-aged WT mice, but not in young WT, young *Catb*^−/−^, and middle-aged *Catb*^−/−^ mice.	2017	[141]
		*P. gingivalis* ATCC33277, Kgp-deficient *P. gingivalis* KDP129	*Cx3cr1^+/GFP^* mice	Injection of *P. gingivalis* into the somatosensory cortex of mice	GFP^+^ microglia accumulated around the injection site of *P. gingivalis*, but not of KDP129.	2017	[137]
		PgLPS	Rats (*n* = 6)	Palatal gingival infection 3 times for 2 weeks	Induced alveolar bone loss and increased serum Aβ levels.	2019	[137,142]
		*P. gingivalis* ATCC33277	Rats Infected for 4 weeks (*n* = 10) Non-infected for 4 weeks (*n* = 10) Infected for 12 weeks (*n* = 10) Non-infected for 12 weeks (*n* = 10)	Intravenous infection 3 times a week for 4 or 12 weeks	Induced tau hyperphosphorylation (pTau181 and pTau231) in the rat hippocampus.	2021	[143]

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
