# Peer review of "Leaky Gum: The Revisited Origin of Systemic Diseases"

_cells, 2022, doi:10.3390/cells11071079_

Round 1

Reviewer 1 Report

The authors aimed to summarize current knowledge of disease-related oral pathogens and contrast the anatomical structures of oral versus gut mucosal layers in the context of microbial leak into human body, embossing the role of oral pathogens in the development of systemic diseases.

The study covers some issues that have been overlooked in other similar topics. The structure of the manuscript appears adequate and well divided in the sections. Moreover, the study is easy to follow, but few issues should be improved. Some of the comments that would improve the overall quality of the study are:

1-) The manuscript needs grammar correction. Please also check typos thorough the text;

2-) Conclusion and Perspectives Section: This paragraph required a general revision to eliminate redundant sentences and to add some "take-home message".

Author Response

We thank the reviewer for the constructive and positive comments.

1) We have gone through the grammar and typo corrections based on British English. 

2) We revised the manuscript to reduce the redundancy and add more take-home messages as well as some more recent reports. Also, we re-organised the multiple references and Table 2 in chronological order.

We hope the revised version will satisfy the reviewer.

Reviewer 2 Report

the authors made a very extensive and clear paper on the correlation between bacterial colonization of the oral.cavity and a large amount of systemic diseases. even if most of this correlations are known in the scientific world, focusing on this aspect can be useful for many clinicians which could misdiagnose the etiology of a specific disease. I really appreciate the efforts of the authors in preparing this paper which is professional and clear. The only suggestion is to dispose the keywords in alphabetical order. The figures are clear and complete, are they original or taken from third parts? If.yes, third parts should give their consense for copyright issues.

Author Response

We thank the reviewer for the constructive and positive comments.

1) We have gone through the grammar and typo corrections based on British English. Also, replaced the keywords in alphabetical order.

2) We revised the manuscript to reduce the redundancy and add more take-home messages as well as some more recent reports. Also, we re-organised the multiple references and Table 2 in chronological order.

3) The figures are original and the photos are de-identified so that direct consent is not required.

We hope the revised version will satisfy the reviewer.